# Identifying the Impacts of Climate Change and Human Activities on Vegetation Cover Changes: A Case Study of the Yangtze River Basin, China

**DOI:** 10.3390/ijerph19106239

**Published:** 2022-05-20

**Authors:** Lang Yi, Ying Sun, Xiao Ouyang, Shaohua Yin

**Affiliations:** 1College of Life Science and Technology, Central South University of Forestry and Technology, Changsha 410004, China; yilang5758@foxmail.com; 2Hunan Institute of Economic Geography, Hunan University of Finance and Economics, Changsha 410205, China; xiao.ouyang@foxmail.com; 3School of Business, Central South University of Forestry and Technology, Changsha 410004, China; yinsh1962@163.com

**Keywords:** NDVI, spatio-temporal dynamic, influencing factors, contribution degree, Yangtze River Basin

## Abstract

The normalized difference vegetation index (NDVI) is a useful indicator to characterize vegetation development and land use which can effectively monitor changes in ecological environments. As an important area for ecological balance and safety in China, understanding the dynamic changes in land cover and vegetation of the Yangtze River Basin would be crucial in developing effective policies and strategies to protect its natural environment while promoting sustainable growth. Based on MODIS-NDVI data and meteorological data from 2000 to 2019, the temporal and spatial distribution of vegetation coverage in the Yangtze River Basin during the past 20 years were characterized, and the impacts of human activities and climate change were quantitatively evaluated. We drew the following research conclusions: (1) From 2000 to 2019, the vegetation cover of the Yangtze River Basin presented a fluctuating inter-annual growth trend. Except for the Taihu Lake sub-basin, the vegetation cover in other sub-basins showed an upward trend. (2) The vegetation cover exhibited a spatial distribution pattern of “high in the middle and low in the east and west”, with the multi-year average value of NDVI being 0.5153. (3) Areas with improved vegetation cover were significantly larger than the areas with degraded foliage. The central region has stronger overall trend of change than the east, and the east is stronger than the west. These vegetation cover changes are largely related to anthropogenic activities. (4) Vegetation cover changes due to precipitation and temperature exhibited significant spatial heterogeneity. While both temperature and precipitation influenced vegetation cover, the temperature was the leading climate factor in the area. (5) Anthropogenic and climate factors jointly promoted the change of vegetation cover in the Yangtze River Basin. Human activities contributed 79.29%, while climate change contributed 20.71%. This study could be used in subsequent studies analyzing the influencing factors affecting long-term vegetation cover in large-scale watersheds.

## 1. Introduction

Over the past few decades, terrestrial ecosystems have undergone tremendous changes. The major factors causing these changes are the effects of climate and anthropogenic factors [1]. As a key factor in maintaining the balance in terrestrial ecosystems, vegetation has an important role in regulating the regional climate and balancing the global carbon cycle [2]. Due to accelerated global warming, vegetation growth has been gradually influenced by climate change and anthropogenic activities [3]. Analyzing changes in vegetation cover is a useful approach to monitoring the regional ecological environment and comprehending the influence of human activities and climatic factors on terrestrial ecosystems [4]. In recent years, large-scale and long-term vegetation monitoring and driving force analysis have become vital components in studying ecosystem changes and formulating environmental policies to react to global climate change [5,6].

The normalized difference vegetation index (NDVI) can effectively reflect the ratio of vegetation photosynthetic radiation [7]. It is widely applied for research on the inter-annual variations of vegetation cover, the spatial-temporal differentiation patterns, and the various driving forces [8]. Natural and human factors combined affect the change of vegetation cover [9]. Natural factors pertain mainly to the effects of precipitation and temperature changes in vegetation cover [10,11]. Zhou et al. [12] found that precipitation is the dominant climatic factor that influences vegetation cover in some high latitude districts of the northern hemisphere. Suzuki et al. [13] concluded that rising temperatures extend the vegetation growth season and promote vegetation productivity. Mohammat et al. [14] found that drought and cold spring cause a significant slowdown in growth rate and reduced greening of land surfaces in Asia.

Human factors also play a major role in influencing land cover and vegetation. Numerous studies have found that anthropogenic factors can have both negative and positive effects on vegetation cover (for example, the middle and upper reaches of the Yangtze River shelterbelt system construction project). Some human activities can significantly improve vegetation cover, while others can lead to serious and long-term vegetation degradation. For instance, Li et al. [15] pointed out that anthropogenic activities can promote the growth of vegetation change rate in a short period of time. Liu et al. [16] indicated that the effects of anthropogenic factors are the key determinants to vegetation cover for the farming-pastoral zone in northern China. Wang et al. [17] indicated that reverting farmlands into forests and other ecological projects can significantly increase vegetation cover. Maeda et al. [18] and Nunes et al. [19] indicated that excessive deforestation and conversion into farmlands, expansion of construction areas and urban agglomerations, and similar anthropogenic activities have significantly reduced vegetation cover, resulting in land degradation, desertification, and soil erosion.

One country that has experienced massive changes in land cover and vegetation in the last few decades is China. Considerable changes in demographics and economic activities, coupled with the effect of climate change, significantly changed the vegetation cover in China [20]. However, because of the different terrain, hydrothermal conditions, vegetation types, and human activities, many of the results in these studies found significant regional heterogeneity in vegetation cover changes [21,22,23]. Previous studies have largely paid close attention to the effects of anthropogenic and climate factors on the interannual changes in vegetation cover [24]. Qualitative analysis was generally employed in early studies, which often resulted in highly subjective research conclusions. It is difficult to avoid the contradictions of spatial autocorrelation and collinearity among factors. On the other hand, the contribution of climate change and human activities to vegetation change cannot be separated.

In recent years, quantitative analysis of factors affecting vegetation cover changes has become more common. However, identifying the influence of climate change and anthropogenic activities on vegetation cover changes has not been fully explored, particularly their long-term and large-scale effects. The residual method is the most widely used model to separate climate change and human activities, which uses the difference between simulated vegetation change and actual vegetation change to estimate the impact of human activities indirectly. In order to fill the current research gaps by identify the impact of anthropogenic and climate factors on vegetation dynamics, this article explores the response characteristics and long-term trends of vegetation cover changes to anthropogenic and natural factors. 

The study area is the Yangtze River Basin, which is an important area for ecological balance and safety in China. In this area, a large number of ecological greening projects, hydropower development, and urbanization have caused tremendous changes in the vegetation coverage of the Yangtze River Basin. Due to the region’s wide cover and diversity in hydrothermal conditions and ecosystem types, the response of vegetation cover would exhibit quantifiable spatial differences [25]. Correlation analysis has been used to qualitatively analyze the causes of vegetation changes in the Yangtze River Basin [26], but long-term vegetation observation and attribution analysis have been insufficient. In particular, there are few quantitative studies on the changes in vegetation cover in the Yangtze River Basin and the temporal and spatial dynamic changes of driving factors due to climate change and human activities. The lack of understanding of the impact of human activities on vegetation may not be able to correctly assess the vegetation situation. In recent years, the spatial difference in the Yangtze River Basin has implemented targeted measures and projects based on particular regional conditions, resulting in significant vegetation change differences within the watershed. 

The major research purpose in this article is: (1) to analyze the spatial distribution characteristics of vegetation cover in the Yangtze River Basin and the factors affecting its change trend; (2) to separate and quantify the impact of anthropogenic and climate factors on the vegetation cover change and identify the drivers of change in the different regions. MODIS-NDVI data, meteorological data, and land use data from 2000 to 2019 were applied to analyze the spatial-temporal change in vegetation and the impact of anthropogenic and climate factors in the Yangtze River Basin in the last 20 years. The results of this study are helpful to evaluate the impact of ecological restoration measures and deepen the research on the relationship between climate change, human activities, and vegetation. 

## 2. Materials and Methods

### 2.1. Study Areas

The Yangtze River Basin comprises the mainstream of the Yangtze River and its tributaries, situated between 90°30′–122°25′ E and 24°30′–35°45′ N (Figure 1). The Yangtze River Basin accounts for 18.8% of China’s total land area, with an area of 1.8 million square kilometers. It originates from the Tanggula Mountains and straddles China’s three major terraces. Flowing from the northwest to the southeast, the river traverses China’s three major economic regions and passes through 19 administrative regions, including Qinghai, Yunnan, Sichuan, Guizhou, Hubei, Jiangsu, and Shanghai et al. The Yangtze River Basin has complex and diverse geomorphic types, diverse climate types, and diverse vegetation types, including alpine vegetation, shrub, grassland, grass, meadow, cultivated vegetation, coniferous forest, broadleaved forest, mixed coniferous and broadleaved forest and swamp. Complex climate and rich natural resources have bred diverse ecosystem types in the Yangtze River Basin [27]. Therefore, the Yangtze River Basin is the ideal region for studying the temporal and spatial evolution of vegetation cover and its driving factors. 

### 2.2. Data Used

MOD13A3 monthly NDVI data were acquired from the EOS/MODIS data product of NASA (https://ladsweb.Modaps.Eosdis.nasa.gov (accessed on 1 June 2021)), with an image acquisition period from 2000 to 2019 and spatial resolution of 1 km. The MODIS ReProjection Tools software was applied in data preprocessing, such as format conversion, projection transformation, and splicing.

The meteorological dataset came from the National Earth System Science Data Center, National Science and Technology Infrastructure of China (http://www.geodata.cn (accessed on 1 June 2021)), which included monthly average precipitation and temperature at 1 km resolution from 2000 to 2019. Vector boundary cropping was employed to obtain the NDVI, temperature, and precipitation datasets for the Yangtze River Basin, and calculated the annual value using the average method. To generate more accurate and objective data, errors caused by the satellite’s geometric field of view, haze, and atmospheric clouds were corrected, and the Savitzky–Golay filtering was adopted to correct NDVI data. The temperature, precipitation, and NDVI data were resampled to 1 km raster pixels. The elevation dataset was generated from the ASTER GDEM (30 m resolution) from the China Geospatial Data Cloud. The 30 m resolution land use data for 2000 and 2018 were obtained from the Institute of Geographic Sciences and Natural Resources Research, Chinese Academy of Sciences.

### 2.3. Methods

(1)Theil-Sen Median is often apllied to trend analysis for long-term series data [28]. The calculated formula is as follows:(1)$$\beta =\mathrm{mean}\left(\frac{x_{i}-x_{j}}{j-i}\right),\forall j>i$$
where *x_j_* and *x_i_* are sequence data. *β* > 0 means increase, and *β* < 0 means decrease.

Mann–Kendal is a nonparametric method that can reveal monotonic trends in time series. The statistical test is:(2)$$S=\sum_{i=1}^{n-1}\sum_{j=i+1}^{n}sign(x_{j}-x_{i})$$
where
(3)$$sign(x_{j}-x_{i})=\left\{\begin{array}{ccc} 1 & & \left(x_{j}-x_{i}>0\right)\\ 0 & & \left(x_{j}-x_{i}=0\right)\\ -1 & & \left(x_{j}-x_{i}<0\right) \end{array}\right.$$

*S* obeys standard normal distribution, variance *Var*(*S*) = *n*(*n* − 1)(2*n* + 5)/18. When *n* > 10, the statistic *Z* value is calculated by the following formula:(4)$$Z=\left\{\begin{array}{ccc} \frac{S-1}{\sqrt{Var(s)}} & & \left(S>0\right)\\ 0 & & \left(S=0\right)\\ \frac{S-1}{\sqrt{Var(s)}} & & \left(S<0\right) \end{array}\right.$$

When the *Z* value is greater than 0, it means that the current time series shows an increasing trend; when it is less than 0, it means that the current time series shows a decreasing trend. If the absolute value of *Z* is greater than 1.65, 1.96, or 2.58, the trend pass through the significance test at the 90%, 95%, or 99% confidence level, respectively. 

The Mann–Kendall test can also be further used for the detection of mutation nodes in time series [29]. The detection statistic *Z* is different from the previous one. By constructing an order sequence:(5)$$S_{k}=\sum_{i=1}^{k}\sum_{j}^{i-1}a_{ij}\;\;(k=2,3,4,\cdots ,n)$$
and
aij=1Xi>Xj0Xi≤Xj 1≤j≤i

The average value, *E*(*S_k_*), and variance, *Var*(*S_k_*), of *S_k_* can be calculated as follows:(6)$$E(S_{k})=n(n+1)/4$$
(7)$$Var(S_{k})=\frac{n(n-1)(2n+5)}{18}$$

Meanwhile, the Mann–Kendall forward statistic (*UF_k_*) and backward statistic (*UB_k_*) are defined as:(8)$$UF_{k}=\left[S_{k}-E(S_{k})\right]/\sqrt{Var(S_{k})},\begin{array}{cc} & \left(K=1,2,3,\cdots ,n\right) \end{array}$$
(9)$$UB_{k}=-\left[S_{k}-E(S_{k})\right]/\sqrt{Var(S_{k})},\begin{array}{cc} & \left(K=1,2,3,\cdots ,n\right) \end{array}$$

If *UF_k_* > 0, the series is increasing, while *UF_k_* < 0 indicates a decreasing series. When the *UF_k_* curve and the *UB_k_* curve intersect within the confidence interval, the intersection time is the mutation point. 

(2)Partial correlation analysis is used to control the linear impact of the third variable while analyzing the correlation between two variables [30]. Before calculating the partial correlation coefficient, the correlation coefficient between the two variables should be calculated first:


(10)
$$r_{xy}=\frac{\sum_{i=1}^{n}\left(x_{i}-\bar{x})(y_{i}-\bar{y}\right)}{\sqrt{\sum_{i=1}^{n}{\left(x_{i}-\bar{x}\right)}^{2}}\sqrt{\sum_{i=1}^{n}{\left(y_{i}-\bar{y}\right)}^{2}}}$$


*r_xy_* means the correlation coefficient between variables *x* and *y*, *n* means the number of samples, and $\bar{x}$ and $\bar{y}$ are the average values of the two variables. The formula for the partial correlation coefficient is:(11)$$r_{xy,z}=\frac{r_{xy}-r_{xz}r_{yz}}{\sqrt{\left(1-r_{xz}^{2}\right)\left(1-r_{yz}^{2}\right)}}$$
where *r_xy,z_* represent the fixed variable *z*, and the partial correlation coefficients *r_xy_*, *r_xz_*, and *r_yz_* of *x* and *y* are the simple correlation coefficients for *x/y*, *x/z*, and *y/z*, respectively. We used the *t*-test with α = 0.05 to test the partial correlation coefficient. 

(3)Natural and anthropogenic factors affect the change of vegetation cover together. The predicted value of NDVI can be estimated by a regression equation between NDVI, temperature, and precipitation. The residual is obtained by the difference between the actual and predicted values observed by remote sensing [31]. It can reflect the impact of human activities on vegetation cover and is widely used in related studies [32,33,34]:(12)$$NDVI_{CC}=a\times P+b\times T+C$$(13)$$NDVI_{HA}=NDVI_{real}-NDVI_{CC}$$
where *NDVI_real_* means the true NDVI value, *NDVI_CC_* means predicted NDVI value, *a* and *b* are the regression coefficients for precipitation and temperature, *C* means the constant term, *P* is precipitation, *T* is temperature, and *NDVI_HA_* is residual. 

In order to verify the predicted data, 5000 sampling points were randomly generated in the Yangtze River Basin, and the NDVI observation data and forecast data were extracted through the sampling points for correlation analysis. The validation results show that the NDVI prediction data are accurate enough (R^2^ = 0.9614) to pass the significance test of 0.01, so the residual can be used to separate the effects of climate change and human activities on vegetation cover change (Figure 2).

(4)*NDVI_CC_* and *NDVI_HA_* indicate the changing NDVI trend due to climate and anthropogenic factors. A positive trend rate means that climate or anthropogenic factors promote vegetation cover, while a negative trend rate indicates a reduction in foliage. Based on previous research results [35], the *NDVI_CC_* and *NDVI_HA_* trends were divided into seven levels (Table 1). Using the values shown in Table 2, the contribution rates of climate and anthropogenic factors to vegetation were then estimated and quantified [36,37].

## 3. Results

### 3.1. Temporal Variation Characteristics

#### 3.1.1. Basin-Wide Scale

From 2000 to 2019, the interannual NDVI change in the Yangtze River Basin showed a fluctuating increasing trend (Figure 3). The highest value was in 2017 (0.5592), while the lowest value was in 2000 (0.4625). The inter-annual growth rate was 0.42%/a, and the correlation coefficient between NDVI and year passed through the significance test at the 1% level (*p* < 0.01). The Mann–Kendall test results showed significant differences in NDVI changes and that the curves of the positive statistical series (UF) and the statistical inverse series (UB) intersected around 2007. 

There were two distinct vegetation growth trends that have occurred in the Yangtze River Basin in the last 20 years. Before 2007, vegetation cover had a significant upward trend, increasing by 0.69%/a (R = 0.8283, *p* < 0.01). The results show that the conservation policies, ecological projects, and restoration measures adopted in the river basin have helped promote foliage growth and forest restoration. However, after 2007, the upward trend decelerated to a growth rate of 0.45%/a (R = 0.8103, *p* < 0.01). During this period (particularly in 2008), the Yangtze River Basin experienced an extremely cold winter season. The unusually cold weather conditions in the basin from 2009 to 2010 were coupled with extreme meteorological events and associated disasters, such as droughts, floods, landslides, mudslides, and typhoons, resulting in the slowdown of vegetation growth in the region. Overall, vegetation cover and the general ecological environment in the Yangtze River Basin have improved significantly for the given study period, largely due to the soil and water conservation policies, ecological projects, and environmental measures adopted in the region. 

#### 3.1.2. Sub-Basin Scale

Research results show that vegetation cover in the Yangtze River sub-basins exhibited upward trends, except for the Taihu Basin (Figure 4). The regional heterogeneity of vegetation cover changes among sub-watersheds is significant. The NDVI values in the central sub-basins were significantly higher than the eastern and western sub-basins. The NDVI in the upper mainstream area (Yibin to Yichang), Wujiang River basin, and Jialing River basin increased significantly by 0.73%/10a, 0.67%/10a, and 0.6%/10a, while vegetation cover in the Taihu basin declined by −0.12%/10a.

### 3.2. Spatial Variation Characteristics of NDVI 

#### 3.2.1. Spatial Distribution Characteristics of NDVI 

Due to differences in vegetation types, land use, and climatic factors, vegetation coverage of the Yangtze River Basin showed significant spatial variations from 2000 to 2019. Based on recommendations from existing studies [38] and the actual conditions of the basin, the vegetation cover was divided into four categories. The mean annual NDVI value was calculated at 0.5153, and the spatial distribution of vegetation was characterized as being “high in the middle and low in the east and west” (Figure 5). As shown in Table 3, the area ratio of high vegetation coverage area (NDVI > 0.6) is 31.84% and were mainly distributed in the western Dongting Lake basin, the northeastern Jialing River basin, Wujiang River basin, the Poyang Lake basin, and the western Han River basin. These regions are largely covered by forests, particularly evergreen broad-leaved forests and deciduous broad-leaved. The area ratio of low vegetation coverage area (NDVI < 0.2) is 5.86% and were mainly distributed in the Jinsha River basin and the upstream of Min-Tuo River basin, especially in the upstream of Jinsha River basin, Southern Qinghai, and Eastern Tibet. These areas are mainly alpine grasslands, meadows, and alpine vegetation, and have relatively low vegetation cover due to natural conditions. The harsh climate restricts vegetation growth. At the same time, overgrazing has accelerated grassland degradation, further depleting the region of its vegetation cover. In addition, the further industrialization, urbanization, and construction of transportation facilities in the densely populated areas along the Yangtze River (e.g., Chongqing, Wuhan, Shanghai, and Hangzhou Bay) have accelerated the losses in vegetation cover in these areas.

#### 3.2.2. Change Trend Characteristics of NDVI

The results of the Theil–Sen median trend analysis and the Mann–Kendall test can effectively characterize the spatial distribution of NDVI changes in the Yangtze River Basin. Using recommendations from previous studies [39], the coupling results were divided into five trend types (Table 4). While the NDVI in most areas indicated a growth trend, the NDVI change trend showed spatial heterogeneity. As shown in Table 4, areas with improved vegetation cover accounted for 83.31%, 5.74% had no significant change, and 10.95% experienced vegetation decrease. 

As shown in Figure 6, regions with degraded vegetation cover were obviously smaller than those with improved vegetation. The rate of growth of NDVI in the central region of the basin was faster than that of the eastern and western regions. In terms of spatial distribution, the areas with severe vegetation degradation were mainly distributed in the western Sichuan Plateau, Yalong River, Tongtian River, and Jinsha River. In the upstream of the Yangtze River, areas with decreased vegetation cover were mainly located in high-altitude zones, with their low temperatures and limited precipitation restricting vegetation growth. Anthropogenic activities such as grazing, logging, reclamation, and tourism development have also exacerbated vegetation degradation in these areas. 

Parts of the Yangtze River Delta continue to experience vegetation degradation. These areas are usually at the stage of rapid urbanization development, characterized by accelerated economic development, rising population density, urban area expansion, large-scale infrastructure construction, and increasing fragmentation. For example, urban agglomeration in the midstream of the Yangtze River and Chengdu–Chongqing urban agglomeration, as well as the central cities of major regions, have declining vegetation indexes. 

The introduction of ecological engineering measures such as the Three River Headwater Region protection and the reconversion of farmlands into forests and grasslands have alleviated vegetation loss upstream of the Yangtze River. Vegetation cover in the midstream of the Yangtze River has significantly promoted, especially in Jiangxi, Hunan, Hubei, Chongqing, QinLing-Daba Mountains, Yunnan-Guizhou Plateau, and the adjacent areas of Jiangnan Hills. These areas are the core of forest rehabilitation, implementing ecological developments and conservation measures, such as the Protection Forest Project in the upstream and midstream of the Yangtze River, the South-to-North Water Transfer Project, and the governance of mountains and lakes in the Poyang Lake basin. National and local environmental protection policies have effectively improved the restoration of vegetation in these areas.

The results indicate that, in general, vegetation cover in the Yangtze River Basin has indicated a significant improvement trend in recent 20 years. This is consistent with the results of the research of Qu et al. [40] on the vegetation cover changes in the Yangtze River Basin. The slope of NDVI change is larger at relatively low-altitude areas and those with more frequent human activities. High-altitude areas (such as the Three River Headwater Region) are affected by human activities in different ways. Some ecological policies and measures such as reconversion of farmlands into grassland have considerably increased vegetation cover, while other human activities such as overgrazing and tourism development have significantly depleted foliage. Anthropogenic interventions in the environment can have positive and negative effects on vegetation cover. 

### 3.3. Relationship between Climate Change and Vegetation Change

The main key climate factors affecting the growth and distribution of vegetation are precipitation and temperature [41]. Pixel-by-pixel partial correlation analysis on NDVI, temperature, and precipitation was conducted for the Yangtze River Basin to objectively understand their impact on vegetation growth.

Vegetation cover changes have been found to be significantly affected by climate change and exhibit strong spatial heterogeneity [42]. The spatial distribution of the partial correlation between precipitation and vegetation cover is exhibited in Figure 7a. The proportion of the area with a positive partial correlation between precipitation and NDVI in the Yangtze River Basin is 58.39%, of which 6.29% had a significant positive correlation (Table 5). These regions are distributed upstream of the Central Sichuan Region in the southern part of the Jialing River Basin, Yunnan-Guizhou Plateau, the central hilly area of Sichuan in the southern Min-Tuo River Basin, Qingnan Plateau in the Jinsha River basin, and Jiuhuashan-Huangshan Mountains in the downstream region. Affected by climate change, the increase in precipitation improves the region’s hydrothermal conditions and promotes vegetation growth. The continued improvements in vegetation cover can effectively improve the soil–water conservation in the region.

The area ratio of negative partial correlation between precipitation and NDVI was 41.54%, of which 37.96% had a significant negative correlation. These regions are distributed in the Luoxiao Mountain Area in Poyang Lake Basin, Jinsha River Basin, the Nanling Mountain Area in Dongting Lake Basin, the western part of Min-Tuo River Basin, the Qinling-Daba Mountain area in the nose of the Hanshui River Basin, and the Taihu Basin. The results show that the spatial distribution of vegetation in the Yangtze River Basin is significantly affected by precipitation.

Temperature is another important climatic factor for vegetation growth. Figure 7b presents the partial correlation between vegetation coverage and temperature in the Yangtze River Basin from 2000 to 2019. The proportion of the region with a positive partial correlation between temperature and vegetation coverage was 84.36%, of which 25.45% had significant positive correlation (Table 5). These regions are distributed in the Wuling Mountain–Xuefeng Mountain area in the middle of the Yangtze River sub-basin, the Nanling–Luoxiao Mountain—Wuyi Mountain area in the east, and the upstream of the Jinsha River in the west. Given that high-altitude areas are sensitive to temperature changes, temperature has become the dominant factor restricting the growth of vegetation. The temperature rise in these high-elevation areas supports the increase in vegetation cover. Previous studies have shown that the average temperature has increased significantly in the Yangtze River Basin since the 1970s. The major extreme temperature index showed a significant upward trend for the warm index and a downward trend for the cold index [43]. Due to better temperature conditions, the vegetation cover in the Yangtze River Basin is on the increase. The proportion of the region with negative partial correlation between temperature and vegetation coverage is 15.57%, of which 15.01% had a significant negative correlation. These areas are mainly situated downstream of the middle of the Jialing River sub-basin, the Jianghan Plain in the east of the Han River sub-basin, the Jinsha River sub-basin, the Taihu sub-basin, and the major provincial capital cities.

### 3.4. Relationship between Human Activities and Vegetation Change 

#### 3.4.1. Driving Forces of NDVI Changes in the Yangtze River Basin

Anthropogenic and climate factors are two key driving factors affecting regional vegetation cover. Among them, the direct influence of climate change is mainly manifested by the changes in temperature and precipitation. In terms of human activities, it is affected by the comprehensive effects of ecological construction, urban expansion, agricultural production, and land use. Figure 8 shows the influence of anthropogenic and climate factors on NDVI in the Yangtze River Basin.

The interaction of those two factors can vary considerably for different regions. In high-altitude areas with comparatively low human activities, climate change mainly influenced the vegetation cover. In contrast, in low-altitude areas with generally high population densities, vegetation cover becomes more dependent on human activities, such as hydro-power development and soil erosion control. The results show that in 22.91% of the Yangtze River Basin, vegetation coverage is not significantly affected by climate change. In 66.40% of the basin, climate change promoted vegetation growth, of which 33.06% had moderate or high promotion effects. These regions are distributed in the Wuyi Mountains in the Poyang Lake Basin, the Nanling Mountains in the Dongting Lake Basin, the Wuling Mountains in the Upper Mainstream Area, the Hengduan Mountains in the Sichuan Basin, and the Jinsha River Basin. In 10.69% of the basin, climate change had an inhibitory effect on vegetation growth, of which 2.30% had a moderate or high inhibitory effect. The areas were located mainly in the Yangtze River Delta, the eastern of the Hanshui River Basin, and the densely populated areas, such as Chengdu, Chongqing, Wuhan, and Changsha.

The contribution rate of anthropogenic activities to vegetation growth in the Yangtze River Basin was 83.45% (Table 6). Compared to climate change, areas where human activities had moderate or high promotion effects accounted for a larger proportion (76.74%). Areas where human activities promoted vegetation growth were located in the central Yangtze River Basin, including the upstream area, the Jialing River Basin, Wujiang River Basin, Han River Basin, and Dongting Lake Basin. These regions comprise key regions for water and soil conservation projects, conversion of farmland to forests, and natural forest protection projects in the Yangtze River Basin. The proportion of the area where anthropogenic activities inhibit the growth of vegetation is 13.5%, which is mainly distributed in the upstream Jinsha River basin, Taihu Lake Basin, Southern Qinghai, Eastern Tibet, and many large cities (e.g., urban agglomeration in the midstream of Yangtze River, Yangtze River Delta region, and Chengdu–Chongqing region). These areas have high human activities, such as tourism development, urban construction, mining, and grazing.

In Figure 9, areas where vegetation growth had been caused by both anthropogenic and climate factors accounted for 73.82% of the Yangtze River Basin. Areas where vegetation growth had been influenced only by climate change comprised 1.55% of the basin and were mainly situated in the upstream of Jinsha River Basin in the eastern region of the Qinghai–Tibet Plateau. Areas where vegetation increase had been affected only by anthropogenic factors accounted for 10.43% of the region and can be found in the Jianghan Plain in the east of the Hanshui River sub-basin and the surrounding areas in the north of the Dongting Lake Basin. In these areas, traditional farming techniques are commonly employed, and human planting activities are quite significant. 

In addition, about 8.58% of the NDVI decrease in the Yangtze River Basin was caused by the double influence of anthropogenic and climate factors. These areas are mainly concentrated in the Yangtze River Delta region, the urban agglomeration in the midstream of the Yangtze River, the Hengduan Mountains in the upstream of Jinsha River, and the Western Sichuan Plateau in Lu-Tuo River Basin. Areas with reduced vegetation caused only by climate factors accounted for 0.73% and had a relatively scattered distribution. Areas with reduced vegetation due to anthropogenic factors comprised 4.90% and were distributed in the alpine meadows of the Qinghai–Tibet Plateau upstream of the Yangtze River Basin. The results suggest that the double impact of anthropogenic and climate factors have been major drivers of vegetation change in the Yangtze River Basin in the past 20 years.

#### 3.4.2. The Relative Contribution of Human Activities and Climate Change to NDVI Change in the Yangtze River Basin

The contributions of anthropogenic and climate factors to NDVI have to separated and analyzed individually (Figure 10). Areas where climate change positively contributed to NDVI change accounted for 84.61% (Table 7). Among them, areas with contribution rates of 0–20% and 20–40% comprised 44.31% and 23.86%, respectively. These areas are mainly in high altitudes where precipitation is abundant and vegetation cover change is highly influenced by temperature, such as the Hengduan Mountains, Wuling Mountains, Yunnan–Guizhou Plateau, and Nanling-Wuyi Mountains. Areas with more than 80% contribution rates made up only 3.16%, and were mainly in the Three-River Headwater Region of Qinghai–Tibet Plateau and Hengduan Mountains. Areas where climate change contributed negatively to vegetation cover change accounted for 15.40% and were mainly in the Guoluo area in the southeast of Qinghai–Tibet Plateau, the northern Dongting Lake sub-basin, and the eastern Han River sub-basin. 

As shown in Figure 10b, the contributions of anthropogenic activities to vegetation cover change in the Yangtze River Basin have been mainly positive, affecting 92.77% of the region. Those with contribution rates of 60~80% and greater than 80% made up 23.86% and 54.74% of the basin, respectively. Areas where human activities had contribution rates above 80% are the Dongting Lake Basin, Han River Basin, Poyang Lake Basin, northern Jialing River Basin, and the Yunnan–Guizhou Plateau in the Jinsha River Basin. 

In 7.24% of the study area, anthropogenic activities had negative contribution rates to vegetation cover change. The contributions of that to vegetation cover increase were greater than those from climate change. Using the actual and the mean change trends of NDVI affected by climate change and anthropogenic activities, the contributions of rates were estimated to be 20.71% and 79.29%, respectively. The results suggest that the anthropogenic aspect is the main factor affecting vegetation cover change in the Yangtze River Basin.

At the sub-basin level, the contribution rates of climate change to vegetation cover change ranged between 0% and 55.48%, and only in upstream Jinsha River did climate change contribute more than 50%. The contribution rates of human activities ranged between 44.52% and 95.88% (Figure 11). In 11 sub-basins, human activities contributed more than 50%, which means that in these regions, the effect of anthropogenic factors on vegetation cover change areas was greater than that of climate change. Among them, the contribution rate of human activities in the Han River Basin, Taihu Lake Basin, and the middle reaches of the main stream area exceeded 90%. The contribution of climatic factors to the change of vegetation cover in this region was relatively low, and the change of vegetation cover was mainly affected by human activities.

To better spatially characterize the effect of anthropogenic factors on vegetation change, the spatial configuration of the effects of anthropogenic activities was determined using the Getis-Ord G* statistics. The result suggests that the contributions of anthropogenic factors can be characterized as being “hot in the east and cold in the west” in the Yangtze River Basin (Figure 12). The cold spots and hot spots were found to be closely related to human activities and climate. The hot spot regions accounted for 55.59% of the total area and it mainly concentrated in the Poyang Lake Basin, Han River Basin, Dongting Lake Basin, mainstream area of midstream, Wujiang River Basin, Jialing River Basin, and the Yunnan–Guizhou Plateau in downstream Jinsha River. These regions have a variety of vegetation types and high vegetation cover and are the main areas for environmental management projects (e.g., conversion of farmlands into forests). The cold spots accounted for 25.48% and are mainly distributed in the Jinsha River Basin in the west of the Yangtze River Basin. Cold spot zones are mostly in high altitude areas, which are mainly impacted by climate factors and limited anthropogenic factors.

Based on the change in land use, the impact of human activities on vegetation cover change was discussed by a lot of scholars. With the intensification of human activities, the significant change in land use has profoundly changed the vegetation coverage. As shown in Table 8, the area of land type change in the Yangtze River Basin from 2000 to 2018 was about 143,978.67 km^2^, accounting for about 3.64% of the area of the Yangtze River Basin. The change in land use mainly includes construction land, cultivated land, and forest land, and it is a process of conversion from cultivated land to construction land and forest land. Among them, the largest area was transferred from cultivated land, with a total of 48,778.15 km^2^, which was mainly converted into forest land (40.89%) and construction land (33.67%). Woodland was the largest area, with a total of 46,498.05 km^2^, mainly from grassland (53.98%) and arable land (42.90%).

We know that land use change has obvious positive and negative effects on vegetation change. On the one hand, the urban development and construction in the basin has destroyed the vegetation coverage. On the other hand, ecological construction projects have improved the vegetation coverage. As can be seen from Figure 13, the transfer of land use types in the regions with negative residual change are mainly from cultivated land to construction land and from unused land to grassland, with an area of 6953.4 and 3971.25 km^2^ respectively. In the region with positive residual change, 19,792.73 km^2^ of farmland was converted to forest land, 17,549.23 km^2^ of forest land was converted to farmland, 13,609.27 km^2^ of forest land was converted to grassland, and 24,136.73 km^2^ of grassland was converted to forest land. In addition, human urban development and construction activities have exacerbated vegetation degradation, mainly in urban agglomerations of major river basins (Figure 14). Arable land and forest land are the main sources of construction land, accounting for 73.65% and 17.89% of the inflow area of construction land, respectively. Large-scale urbanization resulted in the conversion of agricultural land into construction land, which destroyed the surface vegetation around the city and significantly reduced the level of regional vegetation coverage. Therefore, ecological engineering, agricultural development and urban construction are the leading human factors of vegetation change in the Yangtze River Basin.

## 4. Discussion

### 4.1. NDVI Spatiotemporal Changes

During 2000 to 2019, the vegetation index in the Yangtze River Basin had a significant growth trend, indicating that vegetation cover has effectively improved. In particular, the results indicate the NDVI in the central part of the basin ameliorated considerably, consistent with the findings of Qu et al. [40]. At the beginning of the 21st century, large-scale ecological projects were employed in China to promote greening and improve the natural environment [44], such as forest restoration, reconversion of farmlands into forests (grass), development of the shelterbelt in the Yangtze River Basin, and the institution of the national nature reserve. However, the anomalous freezing climate at the beginning of 2008 destroyed large areas of forest vegetation in the Yangtze River Basin. As a result, the growth trend of vegetation in this area has slowed down.

Climate is the main factor affecting the annual fluctuations of vegetation cover in the Yangtze River Basin. In terms of space, climate change can have varying effects on the interannual vegetation cover in different parts of the basin. In general, the effects of climate factors are significantly greater than the upstream part of the Yangtze River Basin than those in the middle and downstream regions. The upstream sections and have simpler vegetation cover, particularly in the upstream. Since majority of the upstream lands are at higher altitudes, have lower population density, and are composed of alpine grasslands, the impact of climate change is significantly stronger than the middle and lower reaches with diverse ecosystems, low elevations, and higher population densities.

### 4.2. Influencing Factors of NDVI Change

Climate-induced near-surface atmospheric warming and rising surface temperature trigger major changes in the water and energy balance of terrestrial ecosystems [41], while precipitation and temperature are considered the two most important climatic factors affecting vegetation dynamics. This study also pointed out that the change of vegetation cover in the Yangtze River Basin was mainly positively correlated with precipitation and temperature, indicating that the regional vegetation cover was greatly affected by climate. Studies have indicated that temperature acts as a major role in the spatial configuration of vegetation cover in the Yangtze River Basin [45], especially in the upstream and high-altitude areas. Higher temperatures in temperate environments prolong the growing season, strengthen photosynthesis, and promote vegetation growth [46], especially in areas with abundant water resources [47]. However, when the temperature rises above the threshold for plant development, soil drying caused by surface evaporation can seriously hinder vegetation growth [45]. Since the Yangtze River Basin has rich resources of water and plant life depends less on rainfall, increasing temperature promotes vegetation growth.

From a dynamic point of view, the response of vegetation cover between precipitation and temperature changes can vary considerably in different regions of the Yangtze River Basin. The correlation between temperature and vegetation cover change is greater than that with precipitation. As the temperature rises, the accumulated temperature effect is significant and supports vegetation growth.

Human activities (such as urbanization and artificial ecological restoration projects) also play a major role in affecting vegetation cover in both positive and negative aspects [16]. For example, when Tong et al. [48] removed the influence of meteorological factors on vegetation changes and analyzed the interference of anthropogenic activities in the Guizhou, Yunnan, and Guangxi regions, they found that ecological engineering had a positive influence on vegetation restoration. However, due to the differences in natural conditions, geological background, disturbance modes, and intensity in different geographical locations, the effects of various environmental projects can have significant spatial differences. In this study, the results show that the overall effect of anthropogenic activities on vegetation cover change in the research region is stronger than that of climate change [49]. Anthropogenic factors play a decisive role, particularly in areas experiencing vegetation cover loss [50]. 

In contrast, the effects of climate showed a significant downward trend, indicating that human activities in the region have continued to increase, causing the impact of natural factors on vegetation to weaken. Since the 1980s, vegetation and land cover in the Yangtze River Basin have undergone a significant transformation, with vegetation cover generally exhibiting an upward trend [40]. However, affected by topography and land use types, the impact of different factors on vegetation coverage has clear spatial features. In terms of spatial arrangement, the vegetation cover growth trend in the low-elevation regions of the middle reaches is better than that of upstream and downstream of the Yangtze River. The former is restricted by natural geographical conditions, while the latter is affected by human activities. In addition, areas with low vegetation cover, which used to be found in mountainous regions (used for agricultural production), are now mainly found in urban agglomeration zones. Urban construction has now become the main factor for vegetation cover reduction [51]. 

### 4.3. Future Research Focus

In this study, precipitation and temperature were used to analyze the correlation between climate change and vegetation cover. Due to the complex nonlinear relationship between vegetation cover and meteorological elements, it is not comprehensive to simply use the multiple linear regression model to fit the linear relationship between vegetation cover, precipitation and temperature together, and simply using residual analysis to quantify the impact of human activities on vegetation cover is not comprehensive. Other environmental factors, such as humidity, evapotranspiration [52], sunshine duration, CO_2_ concentration, nitrogen deposition, and soil properties, can all interact with plant growth and land cover change [53] and have a significant impact on vegetation growth. For example, evapotranspiration is directly related to the dry-wet evolution of climate and the physiological activities of the vegetation ecosystem, and has the potential ability to describe vegetation response to the dry-wet evolution of climate [54]. Subsequent studies will continue to improve the selection of other driving factors. In addition, this study did not consider the lag effect of climate factors on vegetation growth in the growing season, and did not extensively consider the response of vegetation to climate change, so the calculated results would have certain uncertainties. Future research should explore the impact of other climatic factors on vegetation coverage, focusing on determining the regional thresholds of vegetation and non-vegetation, as well as the hysteresis effect of different regional meteorological factors on vegetation coverage, and analyzing the inherent response of vegetation to different climatic factors from a dynamic perspective. At the same time, the results of the analysis will be further validated and investigated using other data. This manuscript uses NDVI to analyze regional vegetation coverage. However, the accuracy of NDVI is limited by saturation under dense vegetation cover and its high sensitivity to canopy background brightness. In contrast, the enhanced vegetation index (EVI), which further calibrates the atmospheric variability of raw data based on NDVI, has a stronger ability to reflect vegetation growth in high coverage areas and to classify vegetation types in sparsely populated areas. Therefore, EVI data can be used to monitor vegetation change in complex areas.

In terms of human activities, the temporal and spatial variation of vegetation cover in the Yangtze River Basin was analyzed regionally in this study. The results show that anthropogenic factors have a considerable effect on the Yangtze River Basin vegetation cover change and indirectly decreased the relative influence of climate factors. However, this article only analyzed from the aspect of land-use change and did not premeditate the effect of particular anthropogenic factors, such as vegetation construction, agricultural technology development, and urban expansion, on plant development and foliage [55].

In addition, the residual trend method is biased in estimating climate impacts. Although the residual analysis method widely used in this field was used to quantitatively evaluate the contributions of climatic factors and non-climatic factors, the selection of meteorological factor data and NDVI data would lead to certain uncertainties in the results, and the more precise relationship between meteorological factors and vegetation cover change needs further discussion [45,50]. When anthropogenic and climatic factors both affect vegetation growth, the residual analysis method tends to overestimate the contribution of climate change and underestimate the impact of human activities. In fact, the artificial factor is also a multi-factor unity, having both positive and negative effects. Therefore, conducting field investigations, refining human activity factors, validating training samples, analyzing the accuracy and reliability of the results, and further quantifying the relative contributions of human factors to vegetation cover changes still require in-depth research in the future. In the follow-up research, anthropogenic and climatic factors should be combined to study the driving factors of plant development and vegetation cover changes, and to analyze the evolutionary relationship between human activities, climate change, and vegetation cover.

## 5. Conclusions

Studying vegetation cover changes in the Yangtze River Basin and quantifying the impact of anthropogenic and climate factors on vegetation cover can help improve the regional ecological environment. This article adopted meteorological data and remote sensing data from 2000 to 2019 and various analytical techniques to explore the spatio-temporal vegetation cover change in the Yangtze River Basin and quantify the contribution degree of anthropogenic and climate factors. We drew the following research conclusions:(1)NDVI of the Yangtze River Basin presents a fluctuating increase trend, the inter-annual growth rate is 0.42%/a (*p* < 0.01), indicating significant improvements in vegetation cover for the study period. At the sub-basin level, the inter-annual variations in vegetation cover in each sub-basin were significant in the last 20 years. The NDVI in the central sub-basin was significantly higher than that in the eastern and western sub-basins. Except for the Taihu sub-basin, the vegetation cover in the other sub-basins showed an upward trend;(2)Most areas of the Yangtze River Basin have good vegetation cover, showing a general spatial pattern of “high in the middle and low in the east and west”. The average annual NDVI value was 0.5153. Areas with improved vegetation cover comprised 83.31%, considerably larger than areas with degraded vegetation cover (10.95%). The central region has stronger overall trend of change than the east, and the east is stronger than the west. These findings may be directly related to population density and intensity of anthropogenic activities;(3)Climatic factors affect the change of vegetation cover in the Yangtze River Basin, and NDVI is positively correlated with temperature and precipitation. The effects of climate factors on vegetation cover changes show pronounced spatial heterogeneity. The promoting effect of temperature on vegetation growth is greater than that of precipitation and is the key factor influencing the interannual variations in vegetation cover;(4)In terms of affecting factors, anthropogenic and climate factors have jointly promoted NDVI change in the Yangtze River Basin and have exhibited significant spatial differences. Anthropogenic factors such as urban construction, agricultural reclamation, and afforestation are the leading factors of vegetation cover change, contributing 79.29%, while climate change has contributed 20.71%. Human activities were found to have mixed effects on regional vegetation cover changes and were the main influencing factor, particularly in areas where the vegetation changes significantly. In areas where ecological protection projects have been implemented, human activities have a positive impact. In areas around urban agglomerations with higher economic development, anthropogenic factors have an obviously negative effect. The findings suggest that the human activities influence should be highly considered when developing environmental and economic policies and land use strategies in the Yangtze River basin.

## Figures and Tables

**Figure 1 ijerph-19-06239-f001:**
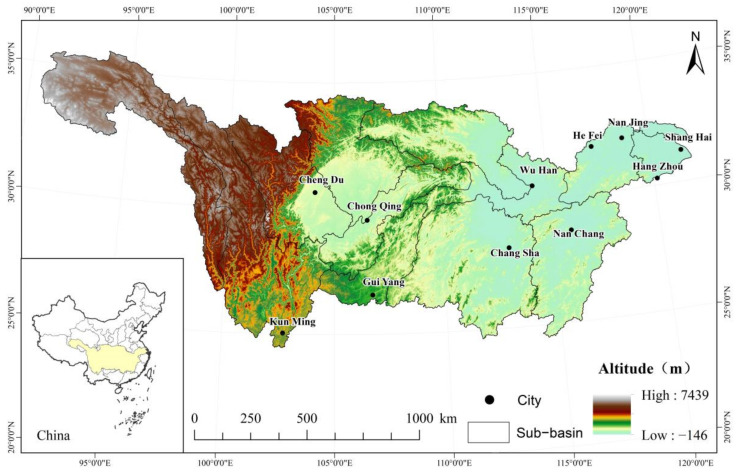
Topographical map of the Yangtze River Basin.

**Figure 2 ijerph-19-06239-f002:**
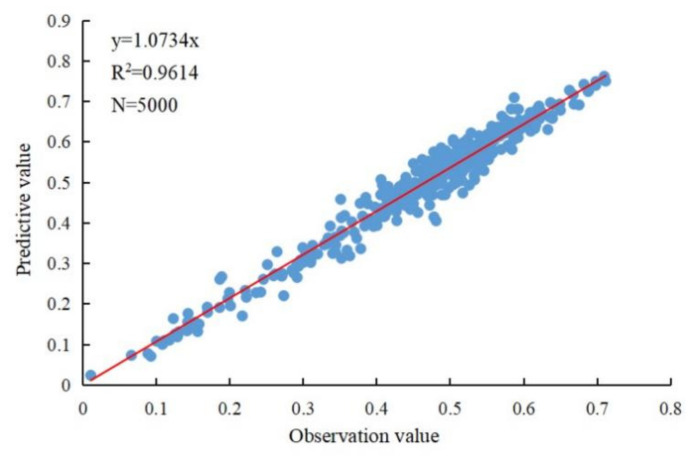
Correlation analysis between observed NDVI and predicted NDVI.

**Figure 3 ijerph-19-06239-f003:**
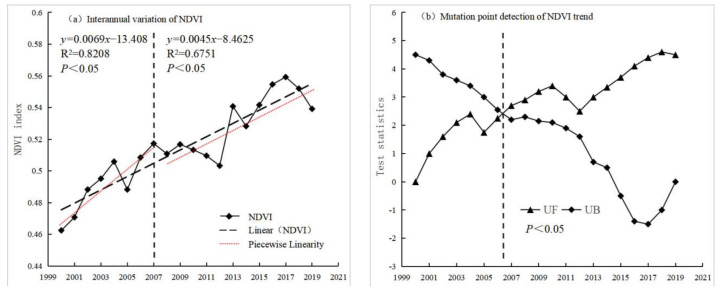
Interannual variation trend and mutation point detection of NDVI of vegetation in the Yangtze River Basin from 2000 to 2019.

**Figure 4 ijerph-19-06239-f004:**
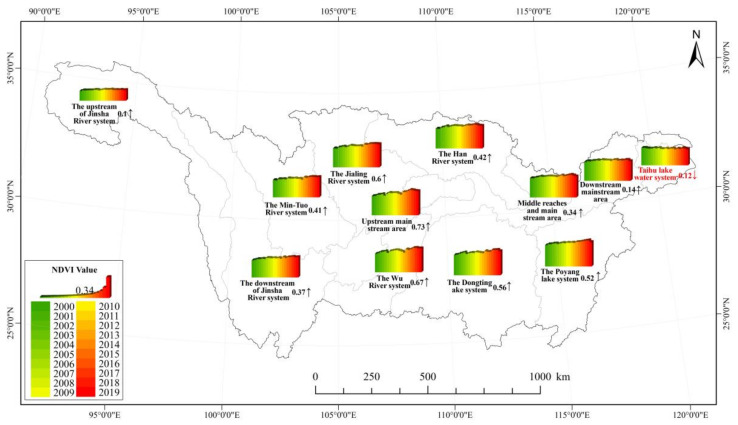
Interannual variation of NDVI in each sub-basin of the Yangtze River Basin (%/a).

**Figure 5 ijerph-19-06239-f005:**
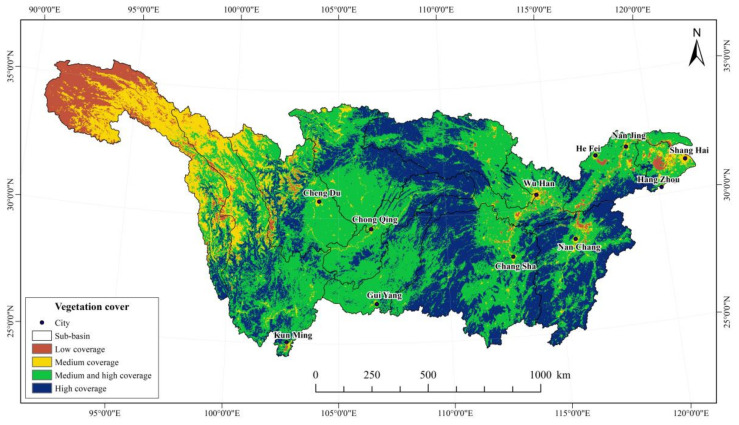
Spatial distribution of vegetation cover in the Yangtze River Basin from 2000 to 2019.

**Figure 6 ijerph-19-06239-f006:**
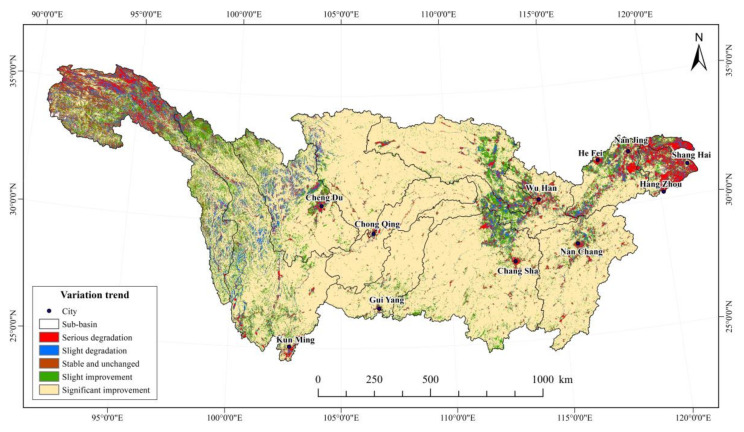
NDVI variation trend in Yangtze River Basin.

**Figure 7 ijerph-19-06239-f007:**
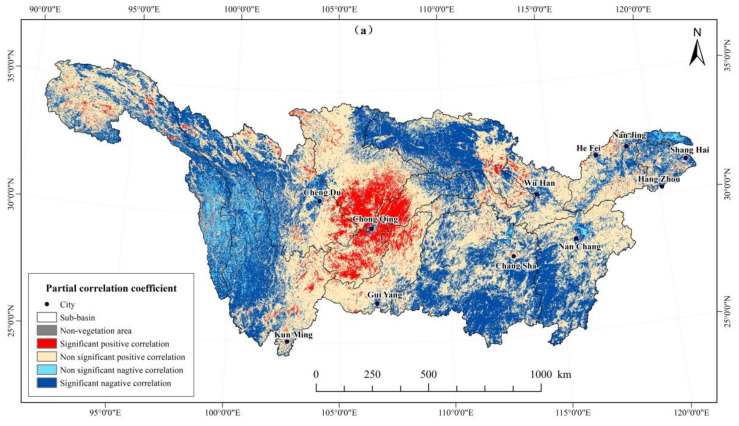
Significance test of the partial correlation coefficient between NDVI with annual average precipitation: (**a**) and annual average temperature; (**b**) in the Yangtze River Basin from 2000 to 2019.

**Figure 8 ijerph-19-06239-f008:**
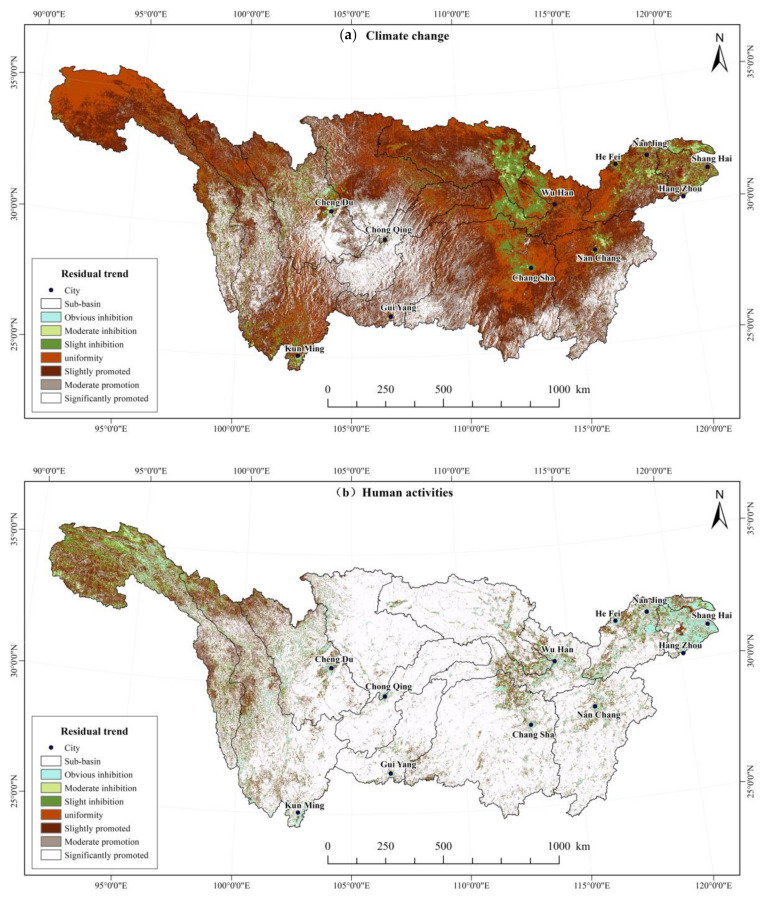
Spatial distribution of impacts of climate change and human activities on vegetation cover change in the Yangtze River Basin from 2000 to 2019.

**Figure 9 ijerph-19-06239-f009:**
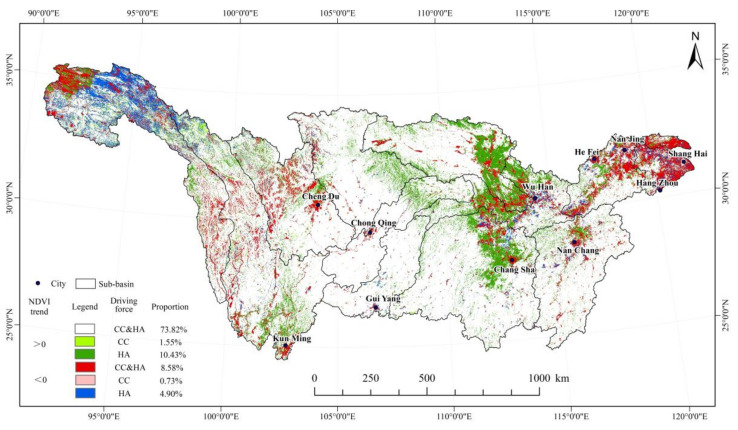
Drivers of vegetation cover change in the Yangtze River Basin from 2000 to 2019 (CC and HA represent climatic change and human activities, respectively).

**Figure 10 ijerph-19-06239-f010:**
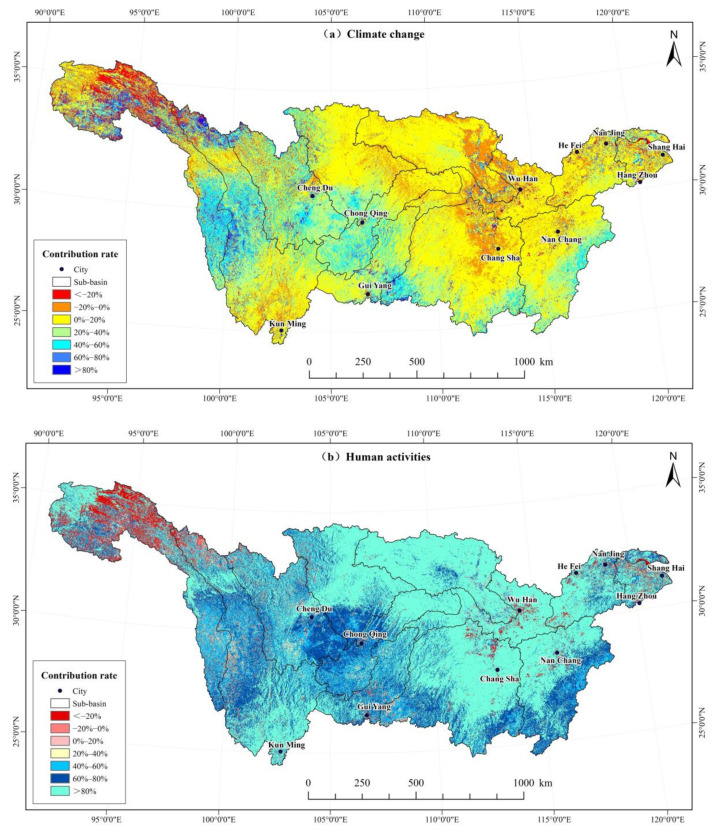
Spatial distribution of contribution rate of climate change and human activities to vegetation cover change in the Yangtze River Basin from 2000 to 2019.

**Figure 11 ijerph-19-06239-f011:**
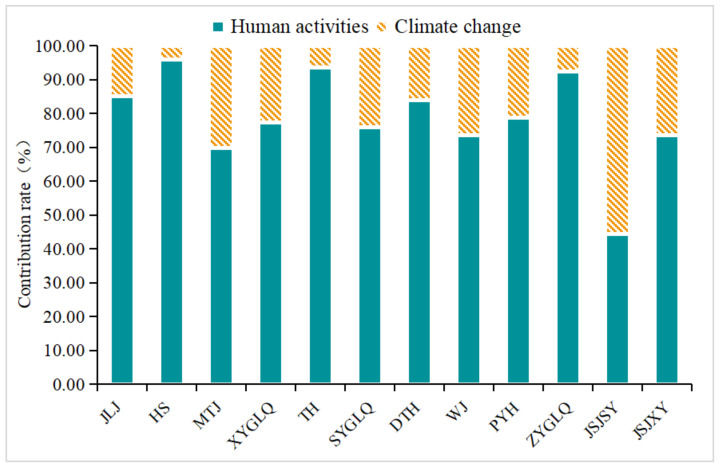
Contribution of climate change and human activities to vegetation cover change in sub-basin from 2000 to 2019. JLJ: Jialing River Basin; HS: Han River Basin; MTJ: Min and Tuo River Basin; XYGLQ: lower main stream area; TH: Taihu Lake Basin; SYGLQ: upper main stream area; DTH: Dongting Lake Basin; WJ: Wu River Basin; PYH: Poyang Lake Basin; ZYGLQ: middle reaches of the main stream; JSJSY: upstream of Jinsha River Basin; JSJXY: downstream of Jinsha River Basin.

**Figure 12 ijerph-19-06239-f012:**
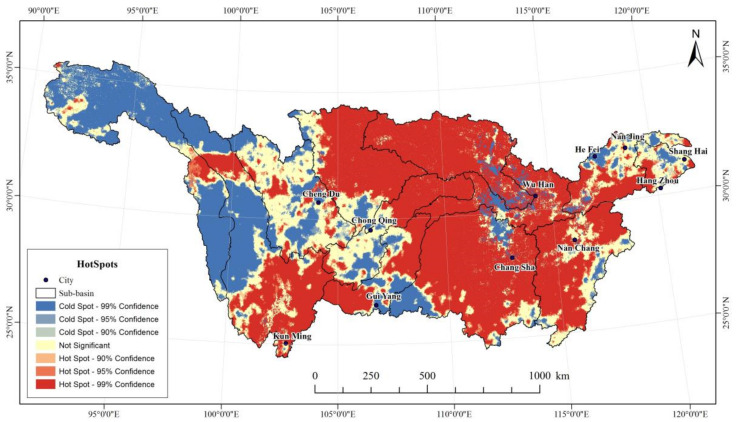
Spatial distribution of cold hot spots of human activity contribution in Yangtze River Basin.

**Figure 13 ijerph-19-06239-f013:**
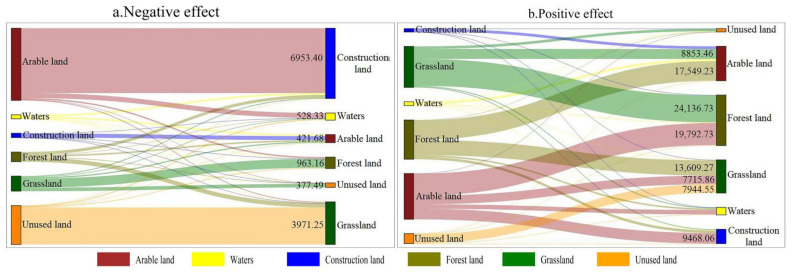
Land use transfer matrix of residual positive and negative effects in the Yangtze River Basin from 2000 to 2018.

**Figure 14 ijerph-19-06239-f014:**
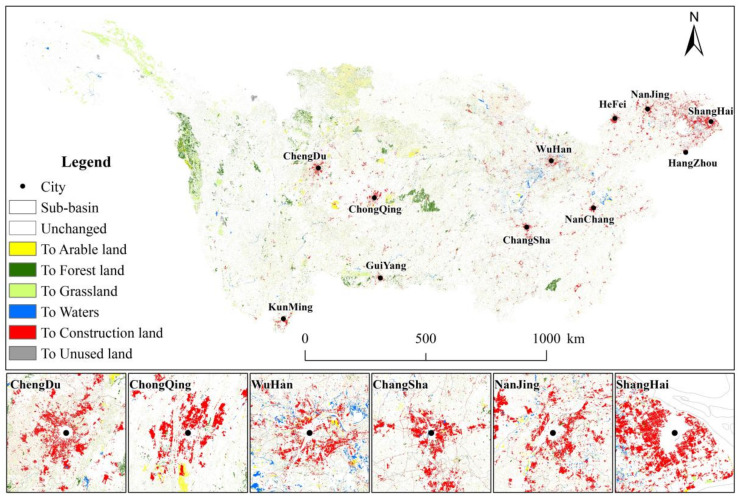
Spatial distribution of land use type change in Yangtze River Basin from 2000 to 2018.

**Table 1 ijerph-19-06239-t001:** Classification of influence of climate change and human activities on vegetation cover change.

Slope (*NDVI_real_*)	Influence Degree
<−2.0	Obvious inhibition
−2.0~−1.0	Moderate inhibition
−1.0~−0.2	Slight inhibition
−0.2~0.2	Uniformity
0.2~1.0	Slightly promoted
1.0~2.0	Moderate promotion
≥2.0	Significantly promoted

**Table 2 ijerph-19-06239-t002:** Criteria for determining the driving factors for the NDVI change and calculation method for the contribution rate.

Slope (*NDVI_real_*)	Driving Factors	Criteria for Dividing the Driving Factors		Contribution Rate of the Driving Factors (%)	
		Slope (*NDVI_CC_*)	Slope (*NDVI_HA_*)	Climate Change	Human Activities
>0	CC & HA	>0	>0	$\frac{\mathrm{Slope}(NDVI_{CC})}{\mathrm{Slope}(NDVI_{real})}$	$\frac{\mathrm{Slope}(NDVI_{HA})}{\mathrm{Slope}(NDVI_{real})}$
	CC	>0	<0	100	0
	HA	<0	<0	0	100
<0	CC & HA	<0	<0	$\frac{\mathrm{Slope}(NDVI_{CC})}{\mathrm{Slope}(NDVI_{real})}$	$\frac{\mathrm{Slope}(NDVI_{HA})}{\mathrm{Slope}(NDVI_{real})}$
	CC	<0	>0	100	0
	HA	>0	<0	0	100

**Table 3 ijerph-19-06239-t003:** Classification standard and data statistics of vegetation cover in the Yangtze River Basin.

Cover	NDVI	Proportion (%)
Low cover	<0.20	5.86%
Medium cover	[0.20, 0.40)	12.20%
Medium and high cover	[0.40, 0.60)	50.10%
High cover	≥0.6	31.84%

**Table 4 ijerph-19-06239-t004:** Statistics of NDVI trend in Yangtze River Basin.

S_NDVI_	Z Value	Trend of NDVI	Percentage (%)
≥0.0005	≥1.96	Significantly improvement	72.64%
≥0.0005	−1.96–1.96	Slight improvement	10.67%
−0.0005–0.0005	−1.96–1.96	Stable and unchanged	5.74%
≤−0.0005	−1.96–1.96	Slightly degradation	6.23%
≤−0.0005	<−1.96	Serious degradation	4.72%

S_NDVI_ is between −0.0005 and 0.0005, and the number of pixels Z > 1.96 or Z < −1.96 is very few, so such pixels are classified as stable invariant type.

**Table 5 ijerph-19-06239-t005:** Correlation statistics of NDVI with precipitation and temperature.

Significant	Significant Positive Correlation	Non-Significant Positive Correlation	Non-Significant Negative Correlation	Significant Negative Correlation	Non-Vegetation Area
Precipitation	6.29%	52.10%	3.58%	37.96%	0.07%
Temperature	25.45%	58.91%	0.56%	15.01%	0.07%

**Table 6 ijerph-19-06239-t006:** Statistics on the influence of climate change and human activities on vegetation cover change in the Yangtze River Basin from 2000 to 2019.

Impact on Vegetation Change	Pixel Scale (%)						
	High Inhibition Effect	Moderate Inhibition Effect	Low Inhibition Effect	Uniformity	Low Promotion Effect	Moderate Promotion Effect	High Promotion Effect
Climate change	0.61%	1.69%	8.39%	22.91%	33.34%	18.29%	14.77%
Human activities	5.51%	3.36%	4.68%	3.01%	6.71%	8.87%	67.87%

**Table 7 ijerph-19-06239-t007:** Statistics of the contribution rates of climate change and human activities to vegetation cover change in the Yangtze River Basin.

Contribution Rate (%)	Pixel Scale (%)						
	<−20%	−20–0%	0–20%	20–40%	40–60%	60–80%	>80%
Climate change	4.97%	10.43%	44.31%	23.86%	10.71%	2.57%	3.16%
Human activities	4.97%	2.27%	0.89%	2.57%	10.71%	23.86%	54.74%

**Table 8 ijerph-19-06239-t008:** Land use transfer matrix in the Yangtze River Basin from 2000 to 2018 (unit: km^2^).

Land Class in 2000	Land Class in 2018						
	Arable Land	Forest Land	Grassland	Waters	Construction Land	Unused Land	Total
Arable land	0	19,947.43	7804.91	4481.13	16,421.46	123.22	48,778.15
Forest land	17,706.62	0	14,072.08	1418.17	2609.78	357.38	36,164.03
Grassland	8975.65	25,099.89	0	867.2	771.99	2668.11	38,382.84
Waters	2097.56	579.13	383.35	0	579.46	337.88	3977.38
Construction land	2867.66	328.9	137.15	214.4	0	10.13	3558.24
Unused land	59.33	542.7	11,915.8	581.35	18.85	0	13,118.03
Total	31,706.82	46,498.05	34,313.29	7562.25	20,401.54	3496.72	143,978.67

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
