# Peer review of "Identifying the Impacts of Climate Change and Human Activities on Vegetation Cover Changes: A Case Study of the Yangtze River Basin, China"

_ijerph, 2022, doi:10.3390/ijerph19106239_

Round 1

Reviewer 1 Report

The review concerns the revised version of the manuscript entitled Identifying the Impacts of Climate Change and Human Activities on Vegetation Cover Changes: A Case Study of the Yangtze River Basin, China.

This version of the article is written appropriately, it is well-organized, contains all of the components I would expect, and the sections are well-developed. The introduction is informative and presents the research topic well.  Materials and Methods are described in detail. The presentation of the results does not raise my objections. The discussion and conclusions are well related to the purpose of the research and the results obtained. The Figures - maps showing the results - are also a strength of the article. References are good and sufficient.

However, there are still editorial errors in this version of the article, most of all there are errors related to the reference to some figures and tables in the text. Below is a list of errors that need to be corrected. These errors are also marked in the attached file.

  • Figures 1, 3, 4, 5, - Unfortunately, there are still no references to these figures in the text;
  • In line 301 – should be Figure 6 instead of Figure 5;
  • Tables 5 and 7 - Unfortunately, there are still no references to these tables in the text
  • Incorrect cite of item 49 in the references

There is

  1. C, X.T.A.B.; B, K.W.A.; B, Y.Y.A.; D, M.B.; E, B.L.; F, C.Z.; C, C.L.A.B.; D, R.F. Quantifying the effectiveness of ecological resto-800 ration projects on long-term vegetation dynamics in the karst regions of Southwest China. Int J Appl Earth Obs 2017, 54, 105-113.

Should be 

Tong, X.; Wang, K.; Yue, Y.;  Brandt, M.;  Liu, B.; Zhang, C.; Liao, C.; Fensholt, R. Quantifying the effectiveness of ecological restoration projects on long-term vegetation dynamics in the karst regions of Southwest China. Int J Appl Earth Obs 2017, 54, 105-113.

  •  See also no 21 in the references.

Best regards,

Reviewer 2 Report

Dear Authors,

You improved the quality of your manuscript at this stage, that result now more clear and well written.

Your analysis to identify the impacts of climate change and human activities on vegetation cover changes are interesting, however, my suggestion is to insert a comparison of vegetation cover and NDVI map, also with Evapotranspiration Map and Aridity Index map. Evapotranspiration and the Aridity index in hydrology and water sciences are two fundamental indicators of climate characteristics that can be easily understood. They are connected between Budyko formulation, which is very easy to implement. Aridity index values  Map can be found in an international database, and so Evapotranspiration can be easily assessed.

I think at this regard you could find interesting information in this open access MDPI paper:

https://www.mdpi.com/2306-5338/8/4/184/htm

My suggestion is to read it (in particular the part that regards aridity index, which indicates also sources for aridity index maps downloading) and use this work as a reference.

Other minor remarks:

In figure 4 and 13, text inside the figure should be plotted bigger.

Please check all the equations dimensions and fonts if they fit with journal template.

Reviewer 3 Report

The paper has improved with the new submission.

Still some issues arise:

  • Define anthropogenic activities and factors in the abstract and propagate throughout. It is necessary to have examples of such activities too.
  • Discuss why the metrics in methods are applied and no others. Parameter selection is the key of this study. Please discuss on parameter selection

  • Include discussion over having better NDVI data for this kind of studies and how the study could be improved

  • Discuss if natural factors can be other rather temperature and rainfall (i.e. wind, etc)

Reviewer 4 Report

Although problems in the choice of methodology persist, because of the use of a vegetation index without taking into account the different types of vegetation existing in the study area, and therefore, different responses to human influence. The authors have substantially improved the manuscript and have argued their methods correctly.

The introduction is now richer and better contextualizes the rest of the manuscript.

The results are adapted and consistent with the methods used, as are the conclusions.

However, some minor corrections are attached:

Line 34: Keywords should not appear in the title.
Figure 1: Please translate the figure completely into English.
Line 137: Please define acronyms the first time they appear in the text to better contextualize and not have to over-understand the acronyms of the models and data acquired.

Author Response

This manuscript is a resubmission of an earlier submission. The following is a list of the peer review reports and author responses from that submission.

Round 1

Reviewer 1 Report

The authors chose an important research question: the driver identification and contribution quantification of natural processes and human activities. However, the method used here is nothing more than a residual analysis. This method has serious potential problems and has application conditions, which have been ignored by many studies. simple analysis by residual method without specific combination of sufficient local real information can lead to very serious misleading. This type of work is nothing more than a combination of the residual methods and data (vegetation index, temperature, precipitation) and can be applied anywhere. Two papers are highlighted again although the first one was included in the reference list.

Zhu, Z., Piao, S., Myneni, R. B., Huang, M., Zeng, Z., Canadell, J. G., ... & Zeng, N. (2016). Greening of the Earth and its drivers. Nature climate change, 6(8), 791-795.

Piao, S., Wang, X., Park, T., Chen, C., Lian, X. U., He, Y., ... & Myneni, R. B. (2020). Characteristics, drivers and feedbacks of global greening. Nature Reviews Earth & Environment, 1(1), 14-27.

Reviewer 2 Report

Dear authors,

First I would like to thank you for presenting your results. In the article, you discussed a very important problem related to the impacts of climate change and human activities on vegetation cover changes in the Yangtze River Basin in the last two decades (2000-2019). I think the research was done very well.  The study design setup and analysis are good, the input dataset is good. The article is understandably written and well-organized, contain all the components I would expect, and the sections are well-developed. The methodology is clearly explained, the results are well described, and the discussion is carried out well. The Figures - maps showing the results - are also a strength of the article. The bibliography is good and sufficient.  However, there are mainly editorial errors in the text, which should be corrected before considering the article for publication. Below is a list of errors that should be corrected. These errors are also marked in the attached file.

  • Figures 1,2,3,4 – Unfortunately, there is no reference to these figures in the text
  • Tables 5 and 7 - Unfortunately, there is no reference to these tables in the text
  • Several parts of the manuscript use a different (larger) font size, these passages are marked in the text, These are the following lines in the text:

288-296;  358; 262-263; 369-370; 598; 602;

Reference: Walker, B.; Steffen, W. The terrestrial biosphere and global change: implications for natural and managed ecosystems. A synthesis of 640 GCTE and related research. 1997. -  no numbering for this reference, should be 1. 

Best regards, 

Reviewer 3 Report

Dear Authors, 

topic of your manuscript is very intestersting, but at this phase is not ready for pubblication. 

My main concern is about the scientific originality and novelty of your methodology. Most interesting part of your work is attributing changes to climate or human activities, but in order to do this you just apply someone else classification (table 2) without giving your personal contribution in term of novelty. 

In addition, some part of the manuscript does not respect the journal layout template, like mathematical formulas or part of the text that appears bigger and with another format, like if they have been copied and pasted (like lines 285-296 and others). It seems you didn't care a lot writing your manuscript properly.

My suggestion is to modify your method including something new and original scientifically speaking, and to write a new manuscript with the correct layout.

Reviewer 4 Report

The paper presents a concrete study of a concrete environmental area in China based on NDVI and other data sources. The study presents concrete results that are valuable. Concrete studies of areas based on remote sensing are missed in the literature and are important. This study is specially ambitious in anthropogenic implications and reflections. They present quantitative results of human-environment factors.

The study projects towards long-term and large-scale effects of the factors discovered in the study (anthropogenic and natural factors)

Causality is explored through correlation analysis of variables including spatial variability.

Authors reflect that long-term data is scarce for further scope of the work. Sub-basin level data is only available in recent years

Data sources are MODIS-NDVI, metereological data and landcover. From them, their drag anthropogenic and natural factors analysis. Major source of analysis is NDVI for temporal and spatial analysis even when MODIS is not high-resolution satellite data. This list does not include DEM.

The paper is strong in data and in statistical tools

The paper is weaker in social data, please discuss

The tool for computing NDVI_HA is sophisticated and valuable as it is residual and it is robust to scarce social data.

Discuss if natural factors can be other rather temperature and rainfall (i.e. wind, etc)

Please, indicate better how landcover data is used and explain it throughout.

Remarks

Define anthropogenic activities and factors in the abstract and propagate throughout. It is necessary to have examples of such activities too.

Specify which data sources are used in the abstract, it is not clear how quantitative the study is.

Explain Savitzky-Golay filtering as it is very relevant and how MODIS is resampled to 1km

Discuss why the metrics in methods are applied and no others. Parameter selection is the key of this study

Include discussion over having better NDVI data for this kind of studies and how the study could be improved

Discuss if natural factors can be other rather temperature and rainfall (i.e. wind, etc)

Reviewer 5 Report

The present manuscript deals with an analysis of the trend in NDVI and the climatic and anthropogenic influences on this vegetation index. The manuscript is easy to follow, although there are quite a few stylistic and formatting errors, citations are not correctly formatted and the language needs revision.

Introduction

The introduction, although well contextualised, lacks a bioclimatic background, as well as a broad description of the vegetation type in the study area. There is no clear starting hypothesis supported by the information in the introduction.

Methodology

The methodology is well described, although I have serious doubts as to whether the methodology is really adequate to achieve the intended objectives. The NDVI index has several drawbacks that have not been addressed and solved by the authors, the most serious of which is that it does not take into account phenology or the period of plant activity, nor is it complemented with other indices to correct other drawbacks. The treatment of the meteorological data is poor and deficient, the number of meteorological stations does not appear, not even the ratio km2/station, nor whether the data are continuous or faulty, probably using a precipitation raster of dubious origin insofar as the links to the web are not available either. This makes it difficult to replicate the experiment in the future.

Results

The results are clear and seem consistent with the methodology employed, a priori they are interesting results, but they do not seem to be contextualised to the characteristics of each vegetation typology. The manuscript argues that there is an increase in NDVI and from this they infer a greater forestation and vegetative growth, but they do not analyse the quality of these forests, or whether they are crops, reforestations...

Discussion

The discussion in some points is contradictory or poorly argued and almost without references to support the discussion of the results.

More specific comments can be found in the attached file.